# Maintaining Inducibility of Dermal Follicle Cells on Silk Fibroin/Sodium Alginate Scaffold for Enhanced Hair Follicle Regeneration

**DOI:** 10.3390/biology10040269

**Published:** 2021-03-26

**Authors:** Kuo Dong, Xinyu Wang, Ying Shen, Yiyu Wang, Binbin Li, Cuiling Cai, Linyi Shen, Yajin Guo

**Affiliations:** 1State Key Laboratory of Advanced Technology for Materials Synthesis and Processing, Wuhan University of Technology, Wuhan 430070, China; 278226@whut.edu.cn (K.D.); sheny3251@163.com (Y.S.); 278196@whut.edu.cn (C.C.); 278100@whut.edu.cn (L.S.); 245636@whut.edu.cn (Y.G.); 2Biomedical Materials and Engineering Research Center of Hubei Province, Wuhan University of Technology, Wuhan 430070, China; 3Hubei Province Research Center of Engineering Technology for Utilization of Botanical Functional Ingredients, Hubei Engineering University, Xiaogan 432000, China; wangyiyu@hbeu.edu.cn

**Keywords:** extracellular matrix, composite scaffold, self-extract, dermal papilla cells, cell-compatibility, hair follicle structure

## Abstract

**Simple Summary:**

Patients with severe skin damage need to transplant artificial skin substitutes to achieve repair, and often the repaired part cannot achieve the regeneration of accessory organs such as hair follicles. It is very important to study artificial skin substitutes that can promote the regeneration of hair follicles. Dermal papilla cells in hair follicles can express a variety of growth factors, which play an important role in the growth of hair follicles. Dermal papilla cells were used as seed cells and cultured with a biomimetic silk fibroin/sodium alginate scaffold similar to the extracellular matrix around the dermal papilla to explore cell-compatibility on the scaffold. Our research found that a silk fibroin/sodium alginate scaffold is a good biomimetic of the extracellular matrix structure, and that the dermal papilla cells could maintain normal morphology and aggregate growth characteristics on the scaffold. We then proved through animal experiments that this scaffold can promote skin repair and induce hair follicle regeneration.

**Abstract:**

The extracellular matrix (ECM) is important for maintaining cell phenotype and promoting cell proliferation and differentiation. In order to better solve the problem of skin appendage regeneration, a combination of mechanical/enzymatic digestion methods was used to self-extract dermal papilla cells (DPCs), which were seeded on silk fibroin/sodium alginate scaffolds as seed cells to evaluate the possibility of skin regeneration/regeneration of accessory organs. Scanning electron microscopy (SEM) graphs showed that the interconnected pores inside the scaffold had a pore diameter in the range of 153–311 μm and a porosity of 41–82%. Immunofluorescence (IF) staining and cell morphological staining proved that the extracted cells were DPCs. The results of a Cell Counting Kit-8 (CCK-8) and Calcein-AM/PI live-dead cell staining showed that the DPCs grew well in the composite scaffold extract. Normal cell morphology and characteristics of aggregation growth were maintained during the 3-day culture, which showed that the silk fibroin/sodium alginate (SF/SA) composite scaffold had good cell-compatibility. Hematoxylin-eosin (H&E) staining of tissue sections further proved that the cells adhered closely and aggregated to the pore wall of the scaffold, and retained the ability to induce differentiation of hair follicles. All these results indicate that, compared with a pure scaffold, the composite scaffold promotes the adhesion and growth of DPCs. We transplanted the SF/SA scaffolds into the back wounds of SD rats, and evaluated the damage model constructed in vivo. The results showed that the scaffold inoculated with DPCs could accelerate the repair of the skin and promote the regeneration of the hair follicle structure.

## 1. Introduction

Burn injuries are a major worldwide public health problem [1] and cause more severe physiological stress than other traumas [2,3]. A three-dimensional (3D) scaffold material is one of the important tissue regeneration materials, because of its special structure, which provides the necessary support and matrix for cell attachment and proliferation, as well as guiding cell differentiation to target functional tissues or organs [4]. Common materials used for 3D scaffolds include collagen, fibrin, elastin, proteoglycans, glycoproteins, and glycosaminoglycans [5]. Skin substitutes represent significant advances in wound care, however, their application has been hindered by high cost, limited effectiveness, and their inability for reconstituting skin appendages [6]. The hair follicle (HF) is an essential accessory appendage of the skin. While repairing wounds, it is of great significance to promote the functional regeneration of HFs. The HF structure includes the epidermis and the dermis [7], made up of hair follicle epidermal stem cells and hair follicle dermal stem cells, respectively. The epidermal part includes the hair matrix, inner root sheath (IRS), and outer root sheath (ORS), and the IRS wraps the hair shaft. And the dermal part includes dermal papilla (DP) and dermal sheath (DS). The formation of HFs depends on signals from the dermis during skin morphogenesis [8,9,10].

With a dense cell structure surrounded by ECM [11], and its own blood supply, dermal papilla cells (DPCs) can regulate a variety of growth factors to cope with various physiological conditions [12,13]. Previous studies have shown that DPCs in hair follicles can induce and promote hair growth [14,15]. Moreover, DPCs regulate the growth and development of hair by releasing a variety of growth factors [16], which are thought to instruct matrix progenitors during hair growth and bulge stem cells during adult hair regeneration in the hair growth cycle [6,17]. Transplanting newborn hair follicles with the upper half removed (including the bulbs) into adult rats can induce new hair formation [18]. Maintaining the potential hair inducibility of DPCs during cell culture is the most important factor for hair follicle morphogenesis and regeneration [19,20]. However, when cells are cultured under traditional two-dimensional (2D) conditions, DPCs tend to lose their hair-inductive capacity [21]. The key to hair follicle reconstruction is that DPCs still have the ability to induce in vitro expansion. N. M. Mali et al. embedded DPCs in an alginate sphere by electrospinning, which upregulated the gene signal of DPCs and maintained the proliferative capacity [22]. Justin J. Y. Tan et al. used varying stiffnesses of poly-ethylene-glycol-diacrylate (PEGDA) to fabricate different microgels to mimic the hair follicular microenvironment for DPC 3D training. The results showed that DPC-related signal expression proteins were easier to express on softer substrates [23]. Skin ECM is composed of a basement membrane (BM) and the extracellular microenvironment of dermal fibroblasts and epidermal keratinocytes, which can support cell adhesion and regulate cell differentiation [24,25]. The interaction between cells and ECM can maintain the undifferentiated state of cells or promote their differentiation by providing a 3D microenvironment [26]. If a structure similar to natural ECM can be provided to ensure the 3D growth of DPCs, it could effectively restore the ability for hair-induction [11].

Silk fibroin (SF) is a natural polymer fibrin with a wide range of applications in biological fields [27], and which can effectively promote the adhesion, growth, and differentiation of different cells in vitro [19,28], with good biocompatibility and degradability [29,30]. However, SF has a poor stability, and certain treatment methods are needed to improve its stability in solution. Bio-inert sodium alginate (SA) is a polysaccharide with a good mechanical strength [31]. A scaffold prepared with SA has good biocompatibility and can promote cell adhesion and growth [32]. Scaffolds are considered to be the best materials for repairing, maintaining, and improving tissue function [33], and providing a suitable platform for cell growth, proliferation, and differentiation. The porous nature of the porous scaffold can help cells complete growth and nutrient exchange, and prevent the formation of necrotic cell centers inside the material [34]. A scaffold made of SF and SA has a loose and porous structure, which can effectively biomimic ECM, and provide a microenvironment for cell migration and proliferation [35]. The ideal porous scaffold has a proper porosity and pore size, and proper cell infiltration to simulate the natural environment of skin growth [36,37]. Inoculating DPCs into the scaffold holes and maintaining the cell proliferation activity and the ability to induce hair follicles is expected to develop a new type of scaffold that can induce the regeneration of damaged skin hair follicle structures.

In this study, we aimed to evaluate the cell-compatibility of a SF/SA composite scaffold based on our previous material research system [38]. DPCs obtained by self-extraction were used as seed cells cultured on the SF/SA composite scaffold, the cell phenotype was identified by staining of specific proteins and cell morphology observation. We also explored the effect of the SF/SA composite scaffold on maintaining the phenotype and aggregation growth characteristics of DPCs. A good cell-compatible scaffold should promote cell growth and proliferation, which is of great significance for the regeneration of hair follicle appendages. Finally, scaffolds with different treatments were transplanted into the back wounds of Sprague Dawley (SD) rats to explore the induction effect on hair follicle regeneration.

## 2. Materials and Methods

### 2.1. Materials

Bombyx mori silk was purchased from Soho Biotechnology Co. Ltd. (Wuhan, China). Sodium alginate, calcium chloride anhydrous, xylene, 100% ethanol, and neutral gum were purchased from Shanghai Sinopharm Chemical Reagent Co. Ltd. (Shanghai, China). 1-ethyl-3-(3-dimethylaminopropyl) carbodiimide (EDC) was purchased from Sigma (USA). Low glucose Dulbecco’s modified eagle medium (DMEM), PBS, trypsin, and penicillin/streptomycin were purchased from HyClone (USA). Fetal bovine serum (FBS) was purchased from Gibco (USA). DAPI and PBS/Tween (PBST) were purchased from Beyotime Biotech Inc (Shanghai, China). The Cell Counting Kit-8 (CCK-8), Calcein-AM/PI live-dead cell staining, and FITC Phalloidin were purchased from YEASEN Biotech Co. Ltd. (Shanghai, China). The Hematoxylin-Eosin Staining Kit, Anti -Cytokeratin 19 Rabbit pAb, Anti-alpha smooth muscle Actin Rabbit pAb actin, and secondary antibody (FITC conjugated Goat Anti-Rabbit IgG) were purchased from Wuhan Servicebio Technology Co., Ltd. (Wuhan, China). Triton X-100 was purchased from Guangzhou saiguo biotech Co., Ltd. (Guangzhou, China). BSA was purchased from F. Hoffmann-La Roche Ltd. (China). Type I collagenase was purchased from Lanjike Technology Co., Ltd. (Shenzhen, China). Chloral hydrate purchased from Shanghai Macklin Biochemical Co., Ltd. (Shanghai, China)

### 2.2. Material Pore Size and Porosity

The preparation of the SF/SA scaffolds followed the research method of our team [38]. Bombyx mori silk was boiled in a weakly alkaline aqueous solution, dried, and stripped, and then dissolved in a ternary solution with a ratio of CaCl_2_, ethanol, and deionized water of 1:2:8, and reacted at 72 °C until it was completely dissolved. After being dialyzed in deionized water for 4 days, a silk fibroin solution was obtained. We adjusted the concentration of the solution to 2% and used EDC as the crosslinking agent, the cross-linked SF solution was obtained after 30 min reaction at low temperature, and the 2 wt% cross-linked SF scaffold was obtained after freeze-drying for 24 h. Ca^2+^ cross-linking was used to improve the stability of the SF scaffold, and the obtained sample was the FCC group. SA was dissolved in deionized water at 60 °C to prepare a 2 wt% SA solution. With the same cross-linking method and the AC group sample was obtained. SF/SA with a mass ratio of 1:1 was reacted under the action of the cross-linking agent EDC to obtain a dermal substitute scaffold, FA. The freezing temperature of each sample was −20 °C. The morphological structure of materials was characterized by SEM (JSM-7500F, Applid Scientific Instruments (Shanghai) Co., Ltd., Shanghai, China).

### 2.3. Isolation and Culture of DPCs

One-month-old Sprague Dawley (SD) rats (40–60 g) (purchased from Hubei Provincial Academy of Preventive Medicine) were sacrificed by cervical dislocation, soaked in 75% alcohol for disinfection and sterilization, and the full-thickness skin on both sides of the tentacles was taken, and the tissue was rinsed with PBS. It was sterilize again with 75% alcohol, rinsed with PBS to remove residual alcohol, and placed in low-temperature low glucose DMEM medium for use. Ophthalmic forceps peeled off the fascia and adipose tissue to obtain a complete hair follicle structure. We separated the hair follicle bulb under a dissecting microscope, and stored it in a low-temperature and low glucose MEM medium. After all the hair follicle bulbs were peeled off, transferred to a sterile petri dish, 3–4 mL of newly prepared 0.2% type I collagenase solution was added, and digested for 2–3 h in a 37 °C, 5% CO_2_ incubator. The tissue was pipetted repeatedly during digestion. Low glucose DMEM complete medium containing 10% FBS and 1% penicillin/streptomycin was added to terminate the digestion, we then centrifuged at 1000 r/min for 5 min, and removed the supernatant. We again added 5 mL complete medium and repeated the centrifugation twice, then 5 mL 20% FBS, 1% penicillin/streptomycin DMEM primary medium was added to resuspend the cells. This was inoculated in a T25 culture flask, the primary culture medium was changed for the first time after 5 days, and then the medium was changed every 2–3 days. When the primary cells were cultured for about 10 days, the cell confluence reached 80%, and the cells were passaged 1:2. We replaced the 20% FBS primary culture medium with 10% FBS for subsequent culture.

### 2.4. Characterization of DPCs with Immunofluorescence (IF) Staining

In order to verify the type of the extracted cells, the cells were stained by IF technology and compared with specific antibody proteins of dermal papilla cells. The cells were incubated for 24 h and then stained with immunofluorescence. We washed the cells twice with PBS preheated at 37 °C and fixed them in 4% paraformaldehyde for 10 min at room temperature. The cells were washed 3 times with PBS for 10 min each time, followed by 0.5% Triton X-100 room temperature permeabilization treatment for 5 min. The cell washing operation was repeated, followed by incubation with 3% BSA for 30 min to complete the serum blocking. We added PBS/Tween (PBST) diluted alpha-SMA and CK19 primary antibodies, respectively, and incubated overnight at 4 °C in a humidified box. After adding the secondary antibody, it was incubated in the dark at room temperature for 1 h. We counterstained the nuclei with DAPI. Specific cell markers were detected under a fluorescence microscope (Olympus, IX71, Shanghai Tonghao Photoelectric Technology Co., Ltd., Shanghai, China).

### 2.5. Characterization Cell Growth Morphology with Cell Microfilaments Staining

The cell microfilaments staining of the extracted cells was performed by FITC Phalloidin, and the growth characteristics and morphology of the cells were observed, as compared with DPCs. The 4% paraformaldehyde fixation and 0.5% Triton X-100 permeation operations were the same as for the IF staining. The FITC Phalloidin working solution was incubated for 30 min at room temperature and protected from light, and washed with PBS 3 times for 5 min each time. We counterstained the nuclei with DAPI. The excitation light sources of 496 nm and 364 nm were selected for fluorescence observation under the fluorescence microscope.

### 2.6. Preparation of Material Extract and Its Biocompatibility Evaluation

According to the extraction ratio of 3 cm^2^/mL (surface area or mass/volume) in the ISO 10993-12 extraction standard [39,40,41,42,43], each sample was cut to obtain 6 circular material pieces, with a thickness of 2 mm and a diameter of 10 mm, which could prepare a total of 3.14 mL of material extraction solution. We placed the material pieces in 75% alcohol for UV sterilization for 4 h. In order to remove excess alcohol, the materials were soaked in PBS buffer for 24 h. We replaced the PBS every 12 h and placed the materials on the reverse side. We dried the materials and placed them in a Teflon test tube for later use. A pipette was used to accurately weigh out 3.14 mL of low glucose DMEM containing 10% FBS, and it was added to the test tube to completely cover the materials. The entire process was completed in a Clean Bench (SW-CJ-2FD, AIRTECH) to ensure that no bacteria invaded. After passing a flame, the test tube was sealed with a sealing glue and placed in a constant temperature shaking incubator (SHA-B) at 37 °C for 24 h to obtain material extracts corresponding to different materials. The extract was filtered through a 0.22 um filter to remove solid residues.

At the same time, after the cells were digested with 0.25% trypsin, the cells were resuspended in complete medium (10% FBS, 1% penicillin/streptomycin in low glucose DMEM). Cells were inoculated and cultured in 96-well plates at a cell density of 1 × 10^4^ cells/mL for 24 h. Each well plate was inoculated with 1 × 10^3^ cells, and three parallel experimental groups were set up for each material extract. The material was extracted after the extraction was filtered with a 0.22 μm filter membrane, replacing the complete medium in the original 96-well plate with material extract culture. The cells were tested for CCK-8 cell proliferation activity on the first, second, and third days. The specific operation was: we aspirated the extract from the original well plate, added 100 μL of low glucose DMEM and 10 μL of CCK-8, and then incubated in an incubator at 37 °C for 2 h in the dark. We measured the OD value at 450 nm using a Thermo LabSystems microplate reader (MK3, MOLECULAR DEVICES (Shanghai) Co., Ltd., Shanghai, China). The value-added dermal papilla cells in the extract of various materials was represented by the OD value.

A Calcein-AM/PI Live-Dead Cell Staining Kit was used for staining DPCs after culturing DPCs in the extract for 1, 2, and 3 days. We aspirated the extract from the well-plate, and diluted the reaction buffer 10 times to obtain 1×Assay Buffer. We added 1 μL of Calcein-AM solution and 3 μL of PI solution to each milliliter of 1×Assay Buffer to obtain a double staining reagent. We added 100 μL of double staining reagent to each well and stained for 15 min in a 37 °C incubator. Fluorescence microscope was used for live/dead cell observation.

The cell morphology was stained and observed on the first, second, and third days of culturing the cells with the material extract, using FITC Phalloidin and DAPI to stain the cell microfilaments and nuclei structure. The staining method was introduced in Section 2.5.

### 2.7. Cell Behavior on the Scaffold

#### 2.7.1. Cell Seeding on the Scaffold

In order to cover the 48-well plate with a well dimeter of 10.2 mm, we used a punch to cut the scaffold to obtain a disk with a diameter of 10 mm. The pretreatment of the material was the same as the preparation of the extract. After soaking in PBS for 24 h, it was dried under UV light for use. Cells were counted after digestion, centrifugation, and resuspension. We adjusted the cell concentration to 1 × 10^5^ cells/mL, then spread the materials in a 48-well plate, and inoculated 100 μL of cell suspension on each material for cell preculture. The material/cell culture system was named M/C. After 24 h of culture, 400 μL of normal medium was added to the well plate. After another 3 days of culture, staining was observed.

#### 2.7.2. Characterization the Behavior of Cells in the Scaffold

We added a certain amount of DAPI staining solution to each well plate covered with materials and covered the sample. They were left at room temperature for 5 min, we then aspirated the DAPI staining solution, and washed the cells twice with PBS for 3 min each time.

Am M/C-culture system was used for hematoxylin-eosin staining. Materials were pre-paraffin embedded as section on an embedding machine (JB-P5, Wuhan Junjie Electronics Co., Ltd, Shanghai, China), then we put the sections in xylene 1, xylene 2 for 20 min, 100% ethanol 1, 100% ethanol 2, 75% alcohol for 5 min, then rinsed them with tap water. The sections deparaffinized to water were stained with hematoxylin staining solution for 5 min, and rinsed with tap water. Then hematoxylin differentiation solution was used to treat the sections, which were then rinsed with tap water again. The sections were treated with hematoxylin Scott tap bluing, and rinsed with tap water. The sections were added to 85%, 95% gradient alcohol, and dehydrated for 5 min. We stained the sections with Eosin dye for 5 min, then put the sections in 100% ethanol 1, 100% ethanol 2, 100% ethanol 3, Xylene 1, and Xylene 2, for 5 min each to complete the section transparency operation. We mounted the slides with neutral gum, observed them under a microscope, and collected and analyzed the images.

### 2.8. Regeneration of Wound Tissue in Experimental SD Rats

#### 2.8.1. Construction of a Full-Thickness Skin Wound Model on the Back of SD Rats

This animal experiment was conducted under the guidance of the Experimental Animal Ethics Committee of Wuhan University of Technology. All surgical operations were performed in a sterilized room. We selected male SD rats weighing 150~200 g, according to the dosage of 0.7 mL/100 g body weight, and used 5% chloral hydrate solution to anesthetize the rats intraperitoneally. The rat’s back hair was removed by a shaver, and we disinfected the wound with iodophor. Two circular areas with a diameter of 12 mm were traced on both sides of the back spine. We used surgical scissors to completely remove the full-thickness skin to create a circular full-thickness skin defect with an area of approximately 113.04 mm^2^, reaching the fascia and stopping bleeding. The experimental SD rats were randomly divided into three groups, without any treatment, covering the SF/SA scaffold without DPCs (blank-material group, B/M), and covering the SF/SA scaffold with DPCs (material-cell group, M/C). Each group of wounds were covered with sterilized petrolatum gauze. We used No. 1 surgical suture thread to suture the wound edge skin at the edge, and then wrapped the wound with sterile gauze, and fixed them with an elastic bandage. The experimental animals were fed in a single cage, which avoided the mutual licking of wounds between animals. We observe the SD rats’ feeding and checked whether the wound scaffolds had fallen off every day.

#### 2.8.2. H&E Staining of Wound Area in SD Rats

The experimental rats were sacrificed at 1 and 3 weeks after the operation to obtain the wound surface. The HE staining method was the same as in Section 2.7.2.

### 2.9. Statistical Analysis

All data are presented as mean ± standard deviation (SEM and OD value). Statistical analysis was performed by one-way ANOVA followed by Tukey’s multiple comparison test using SPSS Statistics 24 (SPSS; Chicago, IL, USA). Differences were considered as significant at *p* < 0.05.

## 3. Results and Discussion

### 3.1. Microstructure Analysis of Scaffolds

Using pure SF, pure SA, and SF/SA mixed solutions, respectively, a simple freeze-drying technique was used to prepare a scaffold with a bionic ECM structure. The macro and microstructure characteristics of scaffold samples are shown in Figure 1. The images of all scaffolds revealed a continuous phase and a porous structure with an interconnected network, and pore diameters varied from 153.091 to 311.305 μm, and the porosity between 41.637 and 82.419% (Table 1). A large number of pores with moderate diameter were obtained due to the freezing temperature of −20 °C during the freeze-drying process, which determined the pore structure of the scaffolds after freeze-drying [44]. The pore diameter distribution in the FCC group (Figure 1a) was irregular, and larger pore diameters were dominant. Probably due to the cross-linking effect of Ca^2+^, the structure was damaged to a certain extent. The AC group (Figure 1b) presented relatively regular pore shapes and the pore diameter distribution was more uniform than that in FCC. As shown in Figure 1c, the composite scaffold showed a uniform pore diameter distribution with smaller pore diameter and more regular pore shapes than that in both the group FCC and AC scaffolds. In addition, there were more secondary pores on the wall of the main pore in the composite scaffold. In Table 1, with the addition of SA, the pore diameter and porosity of the composite scaffold became smaller [45]. The FA scaffold had a uniform pore diameter distribution and regular pores, which may better simulate the ECM structure and provide a more suitable microenvironment for cell adhesion and growth [46].

### 3.2. Identification of Extracted DPCs

Currently, it is believed that DPCs are able to be induced to differentiate to regenerate the HF, however, this ability will degrade as the cell passage increases, with the manifestation of a loss of morphology characteristics and the expression of cell markers [21,47]. Therefore, the DPC should be fresh before it is seeded onto the scaffolds, aiming to ensure and maintain its differentiation capacity. Instead of using the conventional stem cell culture medium, we chose the easy-to-obtain low glucose Dulbecco’s modified eagle medium (DMEM) to cultivate DPCs.

Cytokeratin 19 (CK-19) and alpha smooth muscle actin (alpha-SMA) are commonly expressed proteins in the structure of hair follicles. CK19 is a member of the keratin family, responsible for the structural integrity of epithelial cells, and is specifically expressed in epithelial cells and mesothelial cells. At the end of the hair follicle, the expression of CK-19 protein is restricted to the ORS of the follicle [48]. Alpha-SMA is a hallmark feature that distinguishes myofibroblasts from fibroblasts, which can accelerate wound repair by shrinking the wound edge [49]. Alpha-SMA is highly expressed in dermal papilla cells and is often combined with cell morphology observation to identify dermal papilla cells.

In order to detect the extracted cell types, we used IF stain to stain specific cell markers, and FITC Phalloidin and DAPI to stain cell microfilaments and nuclei. Figure 2a,b shows IF staining images of alpha-SMA and CK-19. As shown in Figure 2a, the self-extracted cells had a strong positive expression of alpha-SMA, which proved that the cells belonged to the myofibroblast cell line, and alpha-SMA was a relatively specific cell marker of DPCs [50]. It can be clearly observed from Figure 2c that the growth of self-extracted cells had the characteristics of aggregation. In the magnified image Figure 2d in the red area in Figure 2c, the morphology of the extracted cells was similar to fibroblasts, with a single cell showing a long spindle shape and a larger cell volume, the maximum length reached 223.077 μm. These results showed the morphological characteristics and specific cell markers of DPCs cultured in vitro. CK-19 protein was a specific marker of epithelial cells, and was often strongly positive in ORS cells (ORSCs) instead of DPCs. Therefore, the negative expression in Figure 2b could also indicate that the self-extracted cells did not belong to the epithelial cell line and the mesothelial cell line. All the results above proved that the cells extracted from the HFs were DPCs. This laid the foundation for further research on materials and cell compatibility.

### 3.3. Evaluation of The Biocompatibility of Materials

#### 3.3.1. Toxicity of Material Extracts to DPCs

As a tissue regeneration material, the most important characteristic is good biocompatibility. In other words, the material ought to be able to maintain or even promote cell proliferation and growth. To explore the effect of the materials on cell proliferation and growth, we used extracts of culture cells, and performed live–dead staining and CCK-8 optical density (OD) value analysis. Figure 3a–d shows the fluorescence micrographs of DPCs stained with a Calcein-AM/PI Live-Dead Cell Staining Kit after being cultured with extracted media of different scaffolds for 3 days. Calcein-AM and PI staining expressed cell survival and apoptosis of cells. Apoptotic cells under a fluorescence microscope were characterized by condensation or fragmentation of nuclei; dead cells appeared red, and normal cells appeared green. It can be seen from Figure 3b that there were fewer surviving cells, and the cells showed the characteristics of apoptotic cells, such as condensation and fragmentation. This may have been due to the fact that there are more Ca^2+^ eluates in the FCC extracted media after Ca^2+^ crosslinking, which inhibit the growth of DPCs, causing the death of a large number of DPCs. Compared with the blank group and the FA group, there were more apoptotic cells in the AC group. It can be seen that the dissolution of SA had a certain adverse effect on the growth of DPCs, but the ratio of live/dead cells was high. The FA group had fewer apoptotic cells due to the decrease in SA content [38]. From Figure 3e, it can be concluded that the FCC group had a serious inhibitory effect on the normal growth and proliferation of DPCs, and the corresponding OD value was the lowest. With the extension of the culture time, this phenomenon did not been improve, and the rate of cell growth was still slow. There was no significant difference for the OD value between the AC group and FA group, and no significant difference between the FA group and blank group. These results indicated that the FA group had the best biocompatibility, and the AC and FA groups could promote the proliferation and growth of DPCs, which provided feasibility to further explore the behaviors of DPSs on the scaffold.

#### 3.3.2. Effects of Material Extracts on the Morphology of DPCs

Figure 4 shows the fluorescence images of the cell microfilaments of DPCs stained with FITC Phalloidin after being cultured in the extracts for 2-day. The cells of the blank group (Figure 4a) maintained the characteristic of aggregation growth, and the cell morphology showed a long spindle shape like normal cells. The FCC group (Figure 4b) had a small quantity of cells, and there was no tendency to aggregate growth. The cells grew slowly and showed an abnormal shape; proving that FCC extracted media had a serious hindering effect on the growth of DPCs. The DPCs cultured in the AC group (Figure 4c) showed no obvious characteristics of aggregate growth, but the cell morphology maintained a long spindle, and the cell growth rate was relatively normal. The FA group (Figure 4d) was the most similar to the blank group, with obvious aggregate growth characteristics, a long spindle morphology as normal DPCs, and the least negative impact on the growth of DPCs, which demonstrated that the FA group formed the most suitable growth environment for DPCs.

### 3.4. Evaluation of Culture between Scaffolds and DPCs

#### 3.4.1. Nuclei Staining in M/C-Culture System

Furthermore, the cell behaviors on scaffolds were verified in vitro. For good tissue regeneration material, being nontoxic is far from sufficient, it also requires the capacity to maintain the characteristics and other biological abilities of seeded cells, and offer a suitable structure for cells to proliferate and migrate.

Before the scaffolds are used in vivo, the interaction between the cells and the scaffolds must be considered [51]. Exploring the interaction between scaffolds and cells through cell proliferation, a fluorescence microscope was utilized to observe the number and behavior of cells on the scaffolds (Figure 5). The fluorescence intensity of the cell nucleus was stronger than the background fluorescence of the scaffold, which provided sufficient contrast for imaging. As the scaffolds had a certain thickness, the fluorescence image showed only the distribution of cells on a certain layer. The distribution of cells along the pore structure can be seen from Figure 5, especially for the AC and FA groups, showing better cellular affinity, which contributed to cell adhesion and proliferation. These results are consistent with Figure 3. The FA composite scaffold had a more regular structure and provided a better environment for cell proliferation and growth.

#### 3.4.2. H&E Staining of M/C-Culture System

Figure 6 shows the H&E staining images of scaffolds with DPCs seeded for 3 days. The nucleus was stained blue, while the cytoplasm and scaffold hole structure were stained purple. From Figure 6a–c, it can be seen that more cells and cell aggregation were found in the FA scaffold (Figure 6c). As the staining was on a two-dimensional plane, the cell morphology of the long spindle could only be seen in a small number of cells. It was difficult to observe DPCs in the FCC scaffold (Figure 6a). There were still aggregated cells in the AC scaffold (Figure 6b), but the number of cells was not as large as in the FA group. These results are consistent with Figure 5. It was worth noting that in the AC group and FA group, the DPCs displayed a certain tendency for clustering, which was the basis for 3D growth in vitro. The clustered areas were especially numerous in the FA group.

From the corresponding enlarged images (Figure 6d–f), it can be seen that the cell shapes in each group were similar, and the nucleus and membrane structure were obvious. Good cell growth morphology and proliferation could be observed in both the AC group and FA group, which were more suitable for the growth of DPCs. These results provided the possibility for further experiments in vivo.

### 3.5. Analysis of Experimental Regenerated Tissues in SD Rats

#### 3.5.1. SD Rat Normal Skin Tissue Section

H&E staining of paraffin sections is one of the commonly used tissue observation techniques. The alkaline hematoxylin stains the chromatin in the nucleus and the ribosomes in the cytoplasm purple, and the acidic eosin stains the cytoplasm and extracellular matrix red. We observed the different tissue structures through staining differences.

The H&E stained image of 150~200 g SD rat normal skin in Figure 7 shows a normal tissue structure. Normal skin has epidermis, dermis, hair follicles, sebaceous gland appendages, and blood vessels, and dermal tissue contains a large number of fibroblasts and elastin fibers, giving the skin elasticity and ductility.

#### 3.5.2. H&E Staining of Newborn Tissue Sections of Damaged Wounds in SD Rats

In order to explore whether the regenerated tissue contained a hair follicle structure, H&E staining was performed on the newborn tissue sections of SD rats.

Figure 8a–c represents the control group without any treatment (Control), the blank SF/SA scaffold group without DPCs (B/M), and the SF/SA scaffold group with DPCs (M/C). One week after the operation, the wound was filled with a large quantity of new capillaries and proliferated fibroblasts, but a complete epidermal tissue was not formed. Due to the foreign body property of the material, the part in contact with the material still had an inflammatory reaction. The B/M group (Figure 8(b1)) showed that the scaffold material was gradually degraded due to cell phagocytosis and tissue fluid infiltration, and a small part of the material may remain in the body, resulting in voids in the tissue. The M/C group (Figure 8(c1)) had the largest number of micro vessels, accompanied by regeneration of some appendages, and the tendency to proliferate and differentiate into new tissue areas. As the DPCs had the ability to promote wound repair, compared with the Control group and the B/M group, M/C showed a more obvious effect of promoting the formation of granulation tissue microvessels [52].

In the third week of skin repair, the number of capillaries in the material group decreased and the lumen was occluded. The epithelial tissue gradually proliferated, and the epidermal regeneration was in a good condition. The inflammatory cells were greatly reduced, the extracellular matrix began to accumulate, and granulation tissue was transformed into fibroblasts and fibrous connective tissue composed of ECM. There were a large number of hair follicles and sebaceous gland structures in the M/C group, which had a tendency to grow into the epidermal layer, similar to a normal tissue section. The above results, taken together, indicated that the SF/SA scaffold inoculated with DPCs can repair skin damage and promote the regeneration of accessory organs such as hair follicles.

## 4. Conclusions

In summary, we prepared different scaffolds with pore diameters that varied from 153 to 311 μm. Compared with pure scaffolds, the FA scaffold cross-linked by EDC had a more regular pore structure and a more uniform pore diameter distribution, which could better simulate porous ECM structure and provide a suitable 3-dimensional growth microenvironment for cells. The extracted DPCs, used as seed cells, were cultured with easy-to-obtain low glucose DMEM. Moreover, the positive expression of alpha-SMA, unique long-spindle cell morphology, aggregate growth characteristics of DPCs, and negative expression of CK-19 proved that the cells obtained from the extraction were DPCs. What is more, the cell live/dead staining kit and CCK-8 showed that the AC group and FA group proved to have the capacity to maintain the normal growth of cells. There was no significance difference of the OD value of CCK-8 between the FA group and the blank group. The DPCs maintained the characteristics of aggregation growth and the normal cell morphology. We stained the cells seeded on the scaffold with DAPI and H&E, and observed good cell adhesion. Although all three scaffolds could provide a 3D growth microenvironment for DPCs, the FA group better simulated the ECM microenvironment around the hair follicle, which promoted the best clustering of DPCs, and it best retained the ability for hair-inducement. We took DPC inoculation or not as a variable, and implanted a SF/SA scaffold treated with different methods into the back wounds of SD rats for in vivo evaluation. The results showed that the scaffold inoculated with DPCs could not only accelerate skin repair, but also regenerated a large number of accessory structures of hair follicles in the dermis.

## Figures and Tables

**Figure 1 biology-10-00269-f001:**
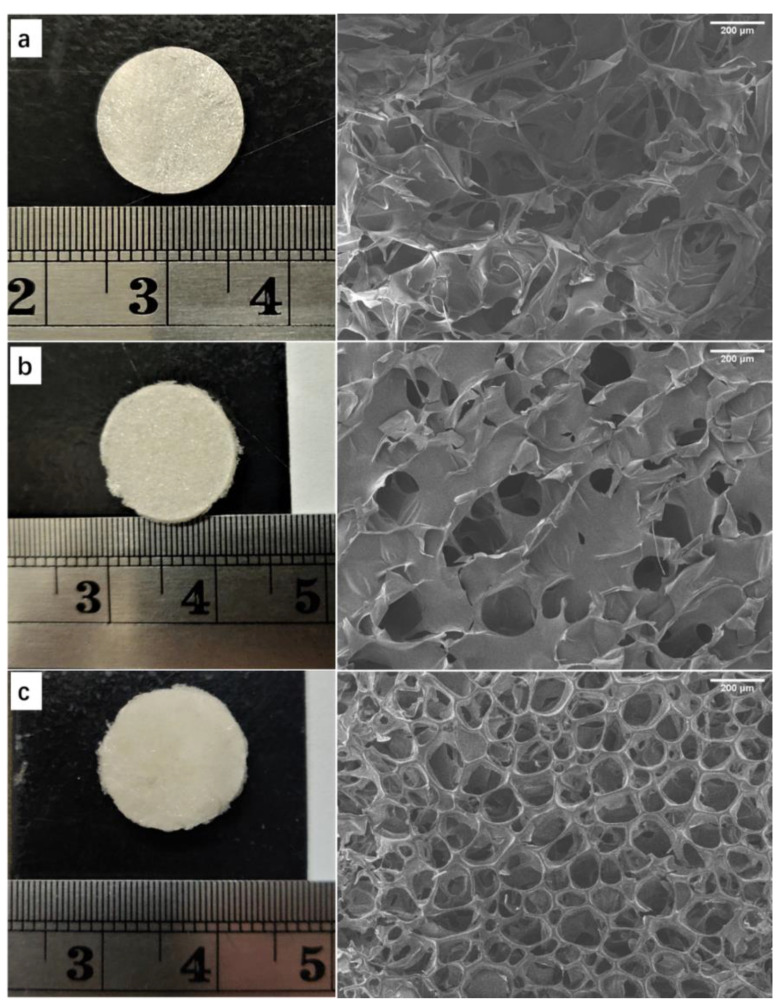
Gross observation and SEM images of different scaffolds. (**a**) FCC, (**b**) AC, and (**c**) FA. Scale bars = 200 μm. The freezing temperature of each sample was −20 °C.

**Figure 2 biology-10-00269-f002:**
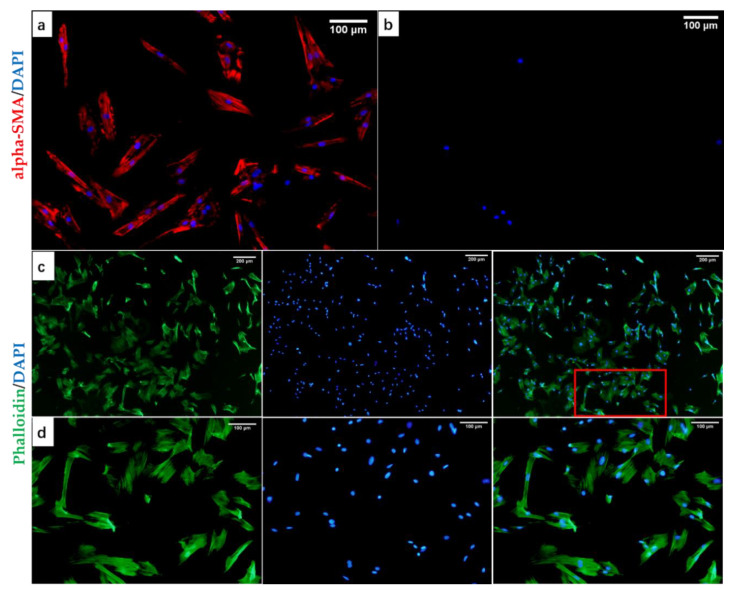
IF staining of dermal papilla cells (DPC) specific proteins and cell morphology. (**a**) Strong positive expression of alpha-SMA (red), (**b**) negative expression of CK-19 (only nuclei expression), (**c**) nuclei/cell microfilaments staining after 3-day of cell culture, including cell microfilaments (green), nuclei (blue), and their combination, (**d**) a partial view of (**c**). Scale bars, (**a**,**b**,**d**) = 100 μm, (**c**) = 200 μm.

**Figure 3 biology-10-00269-f003:**
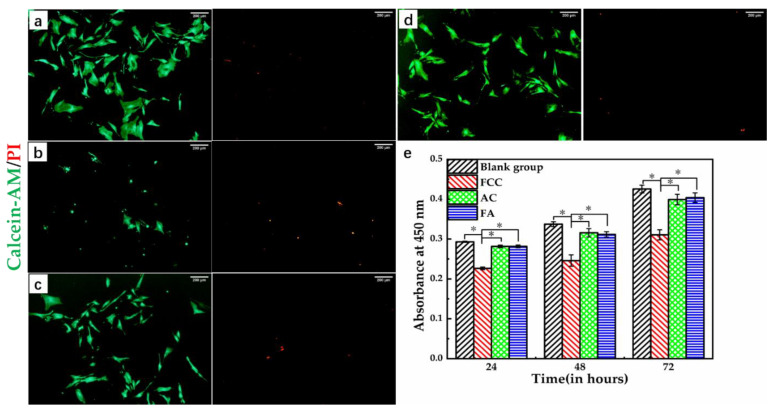
The proliferation and growth of DPCs in the extracted media of different scaffolds. The fluorescent staining images of live-dead cells after 3-day of culturing in: (**a**) blank, (**b**) FCC, (**c**) AC, (**d**) FA extract, respectively. Live cells showed green, and dead cells showed red. (**e**) The cell proliferation after 24, 48, and 72-h culture in the extracted media of different scaffolds tested using CCK-8. * indicates statistically significant differences. (n = 4, per group, *p* < 0.05). Scale bars = 200 μm.

**Figure 4 biology-10-00269-f004:**
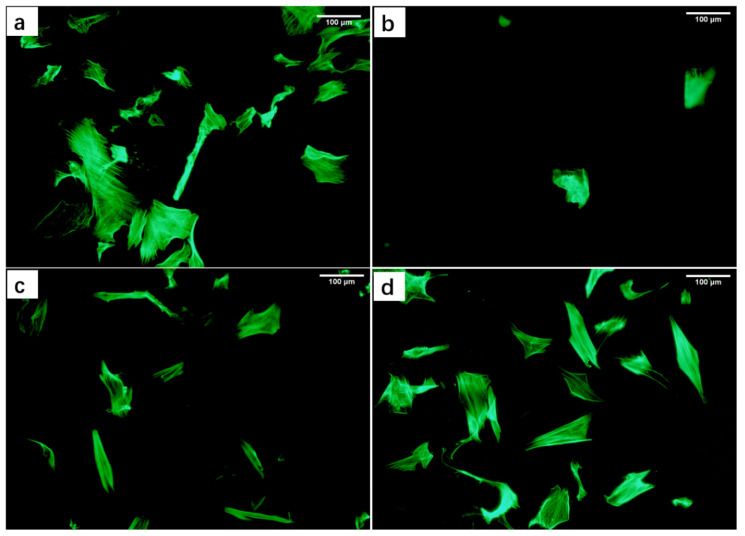
Morphology of DPCs in the extracted media of different scaffolds. Cell microfilaments staining images of DPCs cultured in extract for 2 days. (**a**) Blank group, (**b**) FCC, (**c**) AC, and (**d**) FA. Scale bars = 100 μm.

**Figure 5 biology-10-00269-f005:**
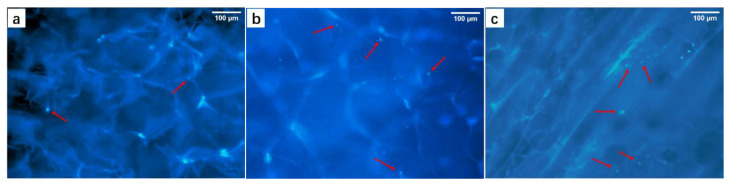
Adhesion of DPCs to the scaffolds. Three days after the DPCs were grown on the scaffolds, the nuclei were stained with DAPI (blue). The arrows point to the cells. (**a**) FCC, (**b**) AC, (**c**) FA. Scale bars = 100 μm.

**Figure 6 biology-10-00269-f006:**
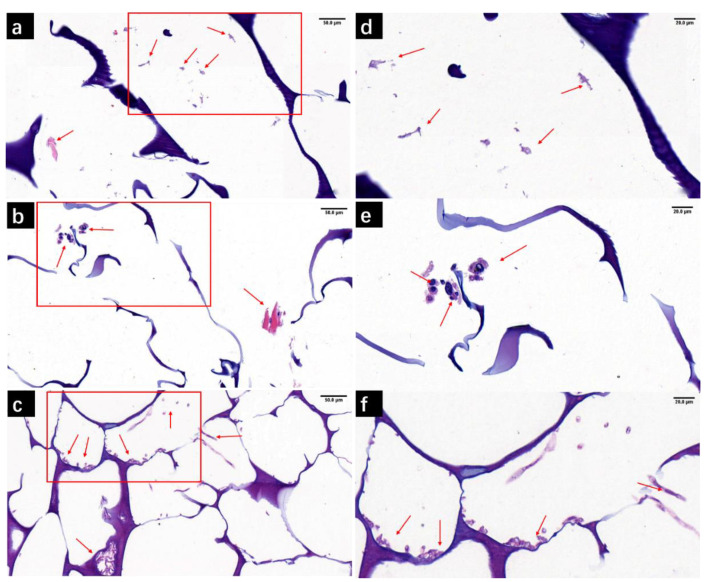
H&E staining showed the distribution of DPCs seeded on the scaffolds for 3-day in vitro culture. (**a**) FCC, (**b**) AC, and (**c**) FA. (**d**–**f**) Corresponding to the partial enlarged diagrams of **a**–**c**. The arrows point mostly to the area where the cells aggregated. Scale bars, (**a**–**c**) = 50 μm, (**d**–**f**) = 20 μm.

**Figure 7 biology-10-00269-f007:**
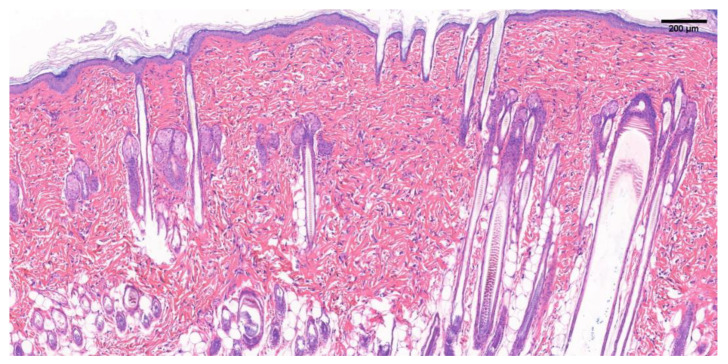
H&E staining image of normal skin tissue of SD rats. Scale bar = 200 μm.

**Figure 8 biology-10-00269-f008:**
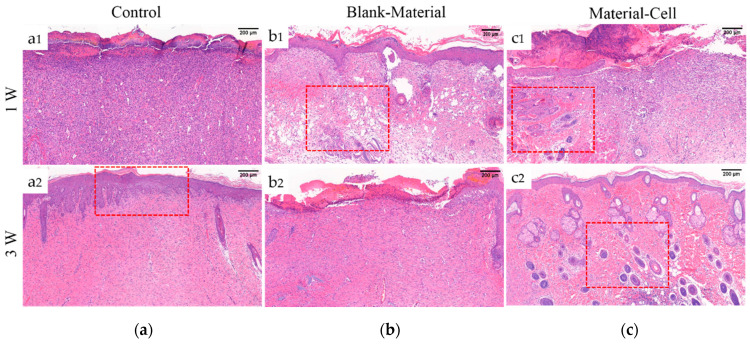
H&E staining of sections of wounds treated by different materials on different days, respectively. (**a**) Control group without any treatment, (**b**) B/M scaffold not inoculated with DPCs, (**c**) M/C scaffold inoculated with DPCs. Scale bar = 200 μm.

**Table 1 biology-10-00269-t001:** Results of porosity and pore size of different scaffolds.

Sample	Porosity (%)	Pore Diameter (μm)
FCC	81.23 ± 1.189	247.028 ± 64.277
AC	43.925 ± 2.288	219.779 ± 35.912
FA	72.845 ± 1.537	173.35 ± 20.259

## Data Availability

Not applicable.

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
