# Peer review of "Maintaining Inducibility of Dermal Follicle Cells on Silk Fibroin/Sodium Alginate Scaffold for Enhanced Hair Follicle Regeneration"

_biology, 2021, doi:10.3390/biology10040269_

Round 1
Reviewer 1 Report
- The morphological characteristics of isolated DPC should be shown in figure.
- In figure 2, only two markers of dermal papilla cells were analyzed, one positive ALPHA SMA and one negative CK19. For better conclusion of dermal papilla cell the authors need to add a couple of more markers.
- In figure 1, The authors have showed only scaffold images. It is recommended to add images of scaffold along with cells seeded on it.
- In figure 3(e), the bar diagram is difficult or confusing to interpret. If number of sample were 4, then why student t-test was performed instead of anova? At 24 hours, there is no difference between FC and AC group then how it shows significance steric? Same at 48 hours? It is recommended to use post hoc analysis for statistical analysis of multiple groups.
- The quality of figure 6 is poor, authors required to present good quality images.
- Uniformity of references is needed. Some references have not mentioned with DOI, journal names or even full name of journals.
Author Response
- Point 1: The morphological characteristics of isolated DPC should be shown in figure.
Response 1: The focus of this thesis is on cell compatibility, highlighting the relationship between cells and materials. Fluorescence staining was performed in the cell identification part to characterize the cell morphology, so the image under the microscope was not added.
- Point 2: In figure 2, only two markers of dermal papilla cells were analyzed, one positive ALPHA SMA and one negative CK19. For better conclusion of dermal papilla cell the authors need to add a couple of more markers.
Response 2: The reason why only alpha-SMA and CK-19 fluorescent labels are made is that the cell extraction area is the hair bulb part of the hair follicle structure, which has limited the cell type to a certain extent. In this study, according to the expression characteristics of different protein markers by dermal papilla cells, characterized from the positive and negative aspects. Under fluorescence photography, it was found that the extracted dermal papilla cells had the same morphology and aggregate growth characteristics as normal cells. Finally, it was concluded that the extracted cells were dermal papilla cells through three aspects.
- Point 3: In figure 1, The authors have showed only scaffold images. It is recommended to add images of scaffold along with cells seeded on it.
Response 3: The basic idea of this article is material preparation, cell extraction, cell identification, material biocompatibility analysis, and cell compatibility analysis. In Figure 1, the gross observation and SEM images of the material are placed in order to better introduce the structure of the scaffold, and the fluorescence image and HE image of the seeded cells are placed in the cell compatibility section. The reason why the SEM image of the cells inoculated with the material is not convenient to express the aggregation and growth characteristics of the cells. At the same time, the high-magnification SEM image of the scaffold material has many wrinkles, similar to the long spindle structure of the dermal papilla cells, which is easy to misunderstand the readers. And the fluorescence staining and HE image can visually show the cell attachment state.
- Point 4: In figure 3(e), the bar diagram is difficult or confusing to interpret. If number of sample were 4, then why student t-test was performed instead of anova? At 24 hours, there is no difference between FC and AC group then how it shows significance steric? Same at 48 hours? It is recommended to use post hoc analysis for statistical analysis of multiple groups.
Response 4: The data was re-edited, anova post hoc analysis was used for significance testing, and the original CCK-8 result graph was modified. There are significant differences between FC and AC groups. Although there was no significant difference between the AC and FA groups, the approximate distribution of cells in space was expressed by HE staining in cytocompatibility.
- Point 5: The quality of figure 6 is poor, authors required to present good quality images.Uniformity of references is needed. Some references have not mentioned with DOI, journal names or even full name of journals.
Response 4: The picture comes from CaseViewer2.4. Due to software limitations, the picture cannot be adjusted for clarity. In order to clearly express the information in the picture, the picture has been enlarged.
The revision of the reference format has been completed.
※Statement: The final version of the article is a comprehensive revision based on the opinions of three reviewers.
Reviewer 2 Report
Please review the microscopy images provided as they require better quality. For example, the fluorescence microscopy images are too bright.
In the conclusion section, mention the round number, for example: 153 to 311 um, instead of 153.091 to 311.305 um.
In the introduction, when mentioning for the first time the polymers, please spell their names completely. For example: Silk Fibroin (SF).... Sodium Alginate (SA). Otherwise the reader will get lost/confused.
Please review the titles of the sub-itens:
2.7.2. Characterization the behaviour of cells on the scaffold
3.4. Evaluation of co-culture between scaffolds and DPCs
In the case of item 3.4, why is it called co-culture? I only identified one culture (DPCs). If so, it is no co-culture because there is no additional cell type in culture.
Please review the English grammar and spelling.
Author Response
- Point 1: Please review the microscopy images provided as they require better quality. For example, the fluorescence microscopy images are too bright.
Response 1: Part of the photos are too bright because they can normally express the information in the pictures. It is difficult to observe a small number of cells in pictures with poor brightness, causing deviations in the results. Some photos have been modified briefly.
- Point 2: In the conclusion section, mention the round number, for example: 153 to 311 um, instead of 153.091 to 311.305 um.
Response 2: The revision of the data in the conclusion has been completed.
- Point 3: In the introduction, when mentioning for the first time the polymers, please spell their names completely. For example: Silk Fibroin (SF).... Sodium Alginate (SA). Otherwise the reader will get lost/confused.
Response 3: The full name supplement in the introduction has been completed.
- Point 4: Please review the titles of the sub-itens:
2.7.2. Characterization the behaviour of cells on the scaffold
3.4. Evaluation of co-culture between scaffolds and DPCs
In the case of item 3.4, why is it called co-culture? I only identified one culture (DPCs). If so, it is no co-culture because there is no additional cell type in culture.
Response 4: The original intention of using the term “co-culture” was that materials and cells were co-cultured. The cells used are single dermal papilla cells rather than multiple cell lines. The material and cell culture system has been redefined in 2.7.1 as the M/C-culture system to replace the co-culture system.
※Statement: The final version of the article is a comprehensive revision based on the opinions of three reviewers.
Reviewer 3 Report
The manuscript deals with the design and in vitro validation of a hybrid construct for skin regeneration, takin g into consideration also hair follicles. Constructs were prepared by blending and crosslinking silk fibroin and sodium alginate, and 3D matrices were obtained by freeze-drying process and characterized n terms of pore size. In vitro evaluations were performed by culturing primary cells isolated from hair follicles of rats. Cells were cultured up to 3 days and characterized with live/dead staining and confocal imaging, cell adhesion morphology, and by histological sections stained with hematoxylin/eosin. The topic is quite interesting in the skin regeneration field, as well as the scientific approach. However, the manuscript is in general not very organized and clear, methods didn’t report enough details, scaffold were not fully characterized and some time the rationale behind the strategies were not clearly stated (ex. why in vitro tests were performed mainly by using extracts and not seeding directly on sponges). English for should be largely revised.
In more detail:
- Introduction: page 2 lines 85-95: it should be re-written because it is not totally clear. For example, how Bio-inert alginate can be biocompatible; why fibroin-alginate scaffolds were considered able to mimic ECM; page 3 lines 99-101: the sentence is not completed, please revise.
- Materials and Methods: 2.2. the method is not clearly described, please revise; scaffold should be characterized at east in terms of blend stability and component releasing, and crosslinking degree; 2.7.1: please explain the rational; 48 well plate doesn’t have a pore size, but well dimeter;
- Results and discussion: 3.1.: what do mean “bionic structure”? why regular porosity is considered biomimetic of ECM and favorable to cell adhesion? Please report a reference; Why cells were not cultured on the scaffolds but just using matrices extracts? 3.3.2 the title is not coherent with the experiment don by using materials extracts; Extracts should be characterized, figure 1 a-b-c it is not essential; Figure 4: resolution should be improved; cells cultured on sponges were visible just in the histological sections, but the magnification was too high, so it was difficult to appreciate the migration into the sponges.
- Conclusions: data included cannot support the conclusions.
Author Response
- Point 1: Introduction: page 2 lines 85-95: it should be re-written because it is not totally clear. For example, how Bio-inert alginate can be biocompatible; why fibroin-alginate scaffolds were considered able to mimic ECM; page 3 lines 99-101: the sentence is not completed, please revise.
Response 1: The modification of page 2 lines 85-95 and page 3 lines 99-101 has been completed.
- Point 2: Materials and Methods: 2.2. the method is not clearly described, please revise; scaffold should be characterized at east in terms of blend stability and component releasing, and crosslinking degree; 2.7.1: please explain the rational; 48 well plate doesn’t have a pore size, but well dimeter;
Response 2: The reason why the preparation of the scaffold has not been introduced in detail is that the preparation method is the research result of our research group, which has been mentioned in the previously published article (References have been introduced in this article) 。
The preparation method has been improved and modified.
- Point 3: Results and discussion: 3.1.: what do mean “bionic structure”? why regular porosity is considered biomimetic of ECM and favorable to cell adhesion? Please report a reference; Why cells were not cultured on the scaffolds but just using matrices extracts? 3.3.2 the title is not coherent with the experiment don by using materials extracts; Extracts should be characterized, figure 1 a-b-c it is not essential; Figure 4: resolution should be improved; cells cultured on sponges were visible just in the histological sections, but the magnification was too high, so it was difficult to appreciate the migration into the sponges.
Response 3:
- What do mean “bionic structure”?
The growth of tissues or cells in vivo requires a specific microenvironment (such as extracellular matrix). This article intends to simulate the microenvironment of the extracellular matrix surrounding DPCs by using scaffolds prepared by SF/SA. Studies have shown that the porous microenvironment is more conducive to cell growth and nutrient transfer (the content has been added to introduction pages 2-3 lines 92-103). The “bionic structure” is the porous structure of the bionic extracellular matrix.
- Why regular porosity is considered biomimetic of ECM and favorable to cell adhesion?
The porosity and pore size of the FA are moderate through the SEM results. This is an idea of the FA group's simulated ECM results. The wording has been revised and the literature has been introduced.
- Why cells were not cultured on the scaffolds but just using matrices extracts?
The extraction method is based on GB, and the results are representative. The reasons for not using the scaffold are as follows. First, during the CCK-8 toxicity test, the presence of the scaffold has a serious impact on the absorbance test, resulting in insufficient data to correctly express the actual conclusion (experimental verification has been carried out). If the scaffold is taken out before the CCK-8 test, it will cause greater errors and make the data meaningless. Second, the same problem exists when staining live/dead cells. The presence of the scaffold causes the overall image to be blurred and cannot correctly express the true result. Therefore, choose the extract instead of the scaffold for the experiment.
- 3.2 The title is not coherent with the experiment don by using materials extracts.
The accuracy of the title has been improved, and the material has been replaced with material extracts.
- Figure 1 a-b-c it is not essential.
The function of Figure 1 a-b-c is to give the reader a visual display.
※Statement: The final version of the article is a comprehensive revision based on the opinions of three reviewers.

Round 2
Reviewer 3 Report
The manuscriot was revised following some of the refrees comments, so improving the scientific impact. However, I'm still considering important to repeat cell culture tests by seeding on scaffold and not extracts. I can understand the technical probem related to the cytotx assay, but for others it should be solved.
Author Response
- Point 1: The manuscript was revised following some of the referees comments, so improving the scientific impact. However, I'm still considering important to repeat cell culture tests by seeding on scaffold and not extracts. I can understand the technical problem related to the cytotx assay, but for others it should be solved.
Response 1:
- Thank you very much for your great suggest. I agree with your pointview that cell culture tests by seeding cells on scaffold is a direct and effective way to evaluate the scaffolds. However, using scaffold extract to evaluate the biocompatibility is also a popular, scientific-proved and effective method. My explains as below:
- We follow the International Standard ( ISO 10993-12) to use the scaffold extract for cell experiment , and the material extraction process is described in detail in the article (page 4,lines 189-203). We also have inserted some articles which use the material extracts to characterize compatibility in recent years (page 4, line 191).
- The preparation method of the SF/SA composite scaffold material involved in this article is derived from the published research results of our group[1]. SF/SA scaffold with a mass ratio of 1:1 has been prepared in the references, and the non-toxicity of the composite scaffold has been proved by Apoptosis and Necrosis Assay Kit (Hoechst and PI) staining and cytotoxicity test.
- In this revision, we added some in vivo test results to further verify the material's non-toxicity and hair follicle inducing ability. Supplements are on page 6, lines 253-274 and pages 13-15, lines 432-472. Due to the addition of part of the in vivo evaluation, on the basis of the previous article, we have revised and supplemented the title, simple summary, abstract and conclusions.

This manuscript is a resubmission of an earlier submission. The following is a list of the peer review reports and author responses from that submission.